# Relationship between Morphological Characteristics and Quality of Aquatic Habitat in Mountain Streams of Slovakia

**Zuzana Štefunková [1], Viliam Macura [1], Andrej Škrinár [1,*] , Peter Ivan [1,2], Milan Čistý [1], Martina Majorošová [1] and Viktória Tyukosová [1]**

[1] Faculty of Civil Engineering, Slovak University of Technology in Bratislava, 810 05 Bratislava, Slovakia; zuzana_stefunkova@stuba.sk (Z.Š.); viliam.macura@stuba.sk (V.M.); peter.ivan@svp.sk (P.I.); milan.cisty@stuba.sk (M.Č.); martina.majorosova@stuba.sk (M.M.); viktoria.tyukosova@stuba.sk (V.T.)

[2] Department of Water Management Operation, Slovak Water Management Enterprise, National Enterprise, 934 80 Levice, Slovakia

\* Correspondence: andrej.skrinar@stuba.sk; Tel.: +421-(2)32-888-617

**Abstract:** This study evaluated the relationship between abiotic flow characteristics and habitat quality. Habitat quality was assessed using the Instream Flow Incremental Methodology (IFIM), which uses bioindication. Brown trout was selected as a bioindicator because of its sensitivity to morphological changes and its occurrence in sufficient reference reaches. The correlation between the morphological characteristics of the stream and the area-weighted suitability (AWS), which represents habitat quality, was evaluated. Fifty-nine reference reaches of fifty-two mountain and piedmont streams in Slovakia were analysed. The correlation analysis demonstrated the strongest relationship between the AWS and the stream depth and width. The relationship between the water surface area and the AWS indicated that, for mountain streams, there is a significantly increasing trend of the AWS value with increasing surface area. Considering piedmont streams, the AWS variation with a change in the water-surface area was minimal. These results can form the basis for deriving regression equations to determine habitat quality. Such a procedure can significantly simplify the evaluation of the quality of aquatic habitat, making it much more accessible for design practice.

**Keywords:** river morphology; habitat; Instream Flow Incremental Methodology (IFIM); area-weighted suitability; habitat suitability curves; water depth; brown trout

## 1. Introduction

The need to address fundamental research questions leading to advances in related science and key management issues has led to the establishment of the scientific discipline ecohydraulics [1] that brings together biologists, ecologists, fluvial geomorphologists, sedimentologists, hydrologists, hydraulic and river engineers, and water resource managers.

A practical example of alternative urban stream channel designs influencing ecohydraulic conditions has been described by Anim et al. [2]. Design scenarios of rehabilitation of the channel morphology were compared against a reference "natural" scenario using ecologically relevant hydraulic metrics. The results showed that: (i) with the addition of natural oscillations to an increasing number of individual topographic variables in a degraded channel, the ecohydraulic conditions were incrementally improved and (ii) the channel reconfiguration reduced the excessive frequency of bed mobility, loss of habitat, and hydraulic diversity, particularly as more topographic variables were added.

The diversity of a physical habitat in rivers and channel connectivity is a necessary requirement for high species diversity and high biotic production [3]. River regulation aimed at flood protection primarily changes the morphology of the riverbed. The diverse natural riverbed has a small capacity; in the case of mountain streams in the Carpathian system, it ranges from $Q_1$ to $Q_5$ (1-year to 5-year floods). Most streams are regulated

to reach a high capacity, usually $Q_{100}$ (100-year flood). Compared with a natural river, a regulated riverbed is monotonous in shape. Such regulated streams do not provide a suitable habitat for a wider range of animals, especially fish. Therefore, it is important to understand the effect of morphology on river ecosystems [4,5]. The geometric properties of the riverbed together with the quality of the habitat form the basis for the assessment of habitat availability [6,7]. The geometry of the riverbed also affects the hydraulic characteristics of the stream, which affects not only the diversity of the velocity field but also the sediment regime, diversity of the riverbed, and a wide range of abiotic and biotic flow characteristics. This relationship can be modelled, which allows a qualitatively higher level for a wide range of water management activities. Habitat modelling in fish ecology [8] provides one of the most comprehensive tools for scrutinizing the river potential to provide a favourable platform for river ecosystems and simultaneously proposes measures to improve the habitat [9]. Instream flow models connect a physical habitat model predicting hydraulic changes to a biological model predicting the response of fish to an altered velocity and depth. Habitat suitability curves (HSCs) based on the frequency of habitat use (fish occurrence relative to available habitat) remain the most widely used biological models in habitat simulations [10].

In one of the most comprehensive studies, Persinger et al. [11] provided a method to study habitat suitability curves on guild levels. Froude number, flow velocity, and water depth were the most important variables for discriminating guilds. However, Gualtieri et al. [12] provided an innovative analysis of 3D spatial habitat metrics based on hydraulic complexity. Guénard et al. [13] highlighted that tidal and hydraulic models coupled with acoustic telemetry and machine learning can be used to predict the spatial distribution of mobile organisms, even in extremely variable ecosystems such as estuaries. Cassan et al. [14] provided a detailed insight into the velocity distributions for modelling the mountain streams for which experimental results from a laboratory-scale model were compared to predictions. Kupferschmidt et al. [15] investigated the effect of the velocity field on various life situations of fish. They showed that habitat use can be monitored even in microhabitat levels, providing a novel video monitoring method that can be easily deployed at remote locations.

Many authors have investigated the influence of hydraulic conditions on various organisms, such as invertebrates [16], macrophytes [17], and even diatoms [18]. However, fish are considered the best bioindicators and are the most sensitive to the morphology of the river channel, including regulated reaches. This has been confirmed by numerous studies [19–21]. The methodology of the Water Framework Directive 2000/60/EC [22] identifies fish as the appropriate bioindicator of morphological changes. This statement applies to streams with good water quality in which continuity has not been interrupted by bed drops, weirs, dams, or similar structures.

The research described in this study is focused on evaluating the relationship between abiotic stream characteristics and habitat quality, as indicated by fish. Based on the many years of our research since 1995, fifty-nine mountain and piedmont streams in Slovakia were topographically surveyed and hydraulically modelled, and from the ichthyological survey of these reaches, the biotic characteristics of the streams were generalized in the form of HSCs. These data were used to evaluate the quality of habitat using the IFIM methodology, specifically the System for Environmental Flow Analysis (SEFA) software [23], as it provides quantitative data that represent the quality of aquatic habitat in the form of area-weighted suitability (AWS). Quantitative evaluation is a particularly important output for communication with water management practice. These models belong to the decision-making family, which means that the obtained data are discussed in the decision-making committee, the composition of which is multidisciplinary. Such an approach is particularly important in the implementation of restoration measures, where the views of water managers, ichthyologists, hydrobiologists, and other experts meet.

The evaluation of aquatic habitat quality by models that provide quantitative data, such as SEFA, requires detailed topographic survey of a representative reach of the riverbed

along with the water surface level and measured flow. The biotic area is represented by the HSCs of individual indicator species. Such modelling requires the interdisciplinary cooperation of experts in the field of hydraulics, hydrology, and ichthyology (assuming the bioindicator is a fish). This makes it time-consuming and practically inaccessible to water management practice. Therefore, it is necessary to look for new procedures that provide equivalent data to the mentioned models. Our prior research [24–26] suggests that the demanding hydroecological modelling could be replaced by regression equations based on the relationship between abiotic instream characteristics and habitat quality represented by fish as a bioindicator. Therefore, the basic aim of this study was to define the relationship between the morphological parameters of natural mountain and piedmont streams and the quality of aquatic habitats.

## 2. Materials and Methods

To achieve the basic aim of the study, the following methodology was chosen:

- Selection of the reference stream reaches;
- Topographic and hydrometric measurements of the reference reaches;
- Ichthyological surveys in the reference reaches of the streams; and
- Correlation and regression analyses of the instream characteristics and their influence on the quality of aquatic habitats.

### 2.1. Selection of Reference Stream Reaches

The degree of reliability of the evaluation of abiotic and biotic characteristics of the aquatic habitat is determined primarily by the size of the file and the homogeneity of the data. A total of 59 suitable localities were selected in mountain and piedmont streams in Slovakia (Figure 1) for the following reasons:

- The negative impact of stream regulation on aquatic biota has been proven predominantly in mountain and piedmont streams.
- These streams are susceptible to morphological changes; therefore, the negative responses of their regulation are common in many areas, and it can be expected that the obtained results can be generalised.
- Mountain and piedmont streams are located in the upper parts of the river basin and are relatively short; therefore, the pollution load of the stream is usually low, and the water quality does not overshadow the influence of riverbed morphology on the quality of the aquatic habitat.

Photos from the reaches taken during field surveys to illustrate the character of the reaches are given in Supplementary Figures S1–S5.

### 2.2. Topographic Survey and Hydrometric Measurement of the Reference Reaches

The hydraulic model was created for all reference reaches of the streams. Therefore, all reaches were surveyed geodetically and characterised by a set of cross sections. The velocity field was modelled directly in the SEFA software [23] (version 1.5, Aquatic Habitat Analysts Inc., Arcata, CA, United States). In the case of a significantly fragmented riverbed (the majority case), the hydraulics was modelled separately in the 1-D HEC-RAS model [27]. Overgrown, hard-to-reach riverbeds were surveyed by cross sections using levelling (Leica Sprinter 150M level machine (Leica Geosystems, Heerbrugg, Switzerland) with an accuracy of ±1.5 mm/km). For more available riverbeds, a Leica FlexLine TS02 total station (Leica Geosystems, Heerbrugg, Switzerland) with an angular accuracy of 3" (1 mgon) and a length accuracy of 1.5 mm + 2 ppm was used. To verify the hydraulic model, fixed geodetic points were established, which allowed us to accurately target the water-level regime at different water conditions by levelling. Measurements were performed in the summer at low discharge.

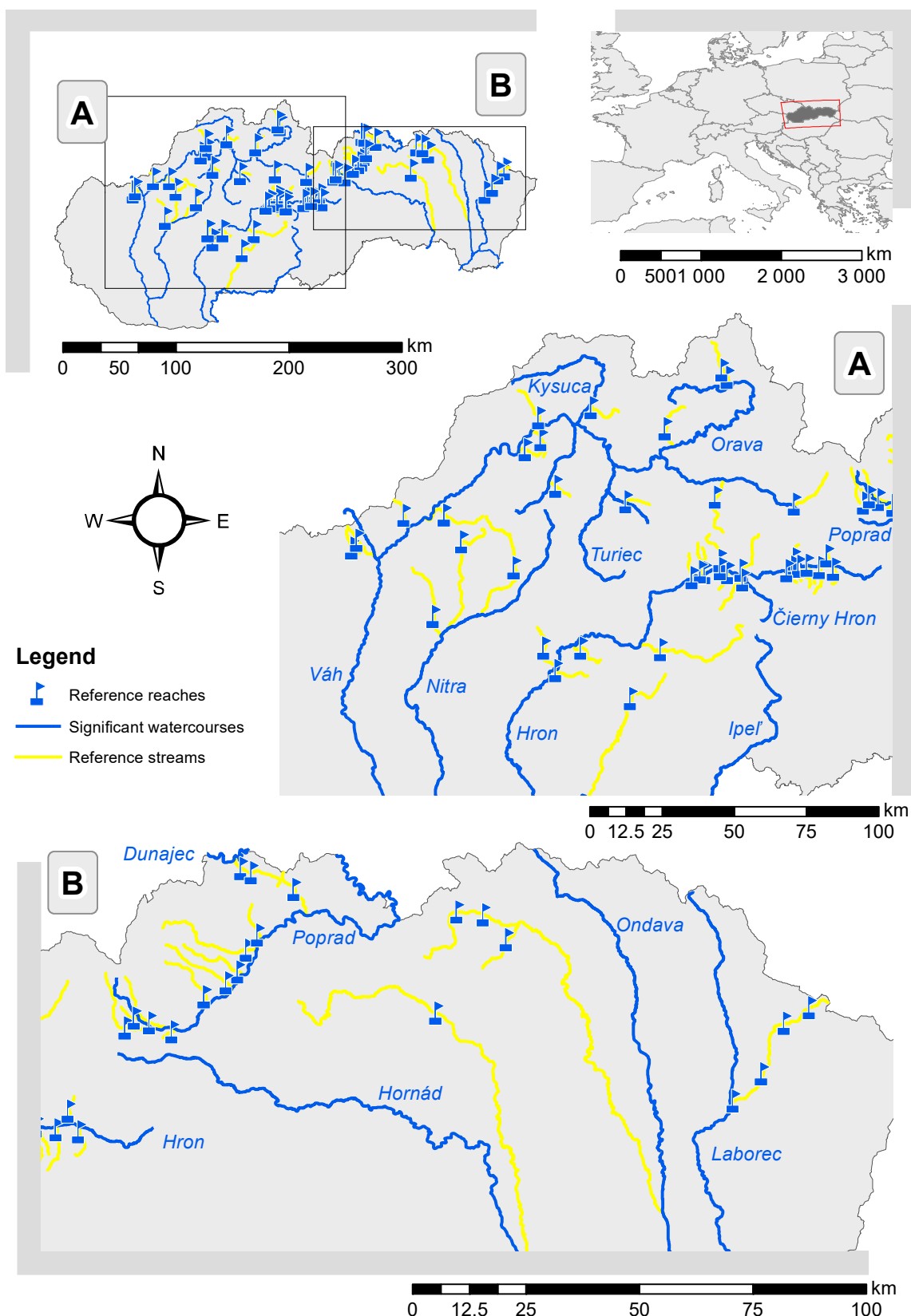

**Figure 1.** Location of reference reaches in streams within Slovakia. (**A**) corresponds to the A cut, and (**B**) corresponds to the B cut from the map scheme of Slovakia.

The discharge determination was performed simultaneously with the ichthyological survey. The hydrometric measurements were performed in accordance with ISO 748 [28]. At the beginning of each measurement, suitable cross sections for hydrometry were selected. The bottom of the riverbed in the measured cross section was regular without large stones or other flow obstacles in the stream, consistent with Herschy [29]. A set of three propellers mounted on one rod was used to measure the flow velocities. All propellers were calibrated according to ISO 3455 [30].

### 2.3. Ichthyological Survey in the Reference Reaches of the Streams

The ichthyological survey aimed to determine the preference for the flow velocities and water depths by individual fish species. Data were obtained by electro-fishing, similar to that reported by Lamouroux et al. [31]. To gather the fish samples, an electro-fishing unit (Hans-Grassl ELT62IIHI (HANS GRASSL GmbH, Schönau am Königssee, Germany)) with the option of a continuous choice of electrical parameters was used. Microhabitat characteristics, water depth, and mean vertical velocity were recorded at the point of capture of each fish. The mean vertical velocity was derived from the measurements recorded by a system of three hydrometric propellers placed on one rod. The location of the hydrometric propellers was standard in the following multiples of water depths (d): $0.2 \times d$, $0.4 \times d$, and $0.8 \times d$.

Fish are good bioindicators that are sensitive to changes in riverbed morphology [32–36] and temperature [37]. For quality assessment and changes in aquatic habitat, one fish species, the adult brown trout (*Salmo trutta m. fario*), was selected using statistical methods. Previous studies [24] stated that trout can sensitively indicate changes in the morphology of the riverbed. The results show that there is a relationship between the morphological parameters of the watercourse and the characteristics of brown trout as a bioindicator of the quality of the aquatic habitat. Brown trout were also present in sufficient quantities in the reference reaches.

From the ichthyological survey, HSCs for individual fish species were derived for each stream. The set of all measured suitability curves of brown trout was generalised for four categories of mountain streams (Figure 2). The generalisation procedure is described in more detail by Macura et al. [25].

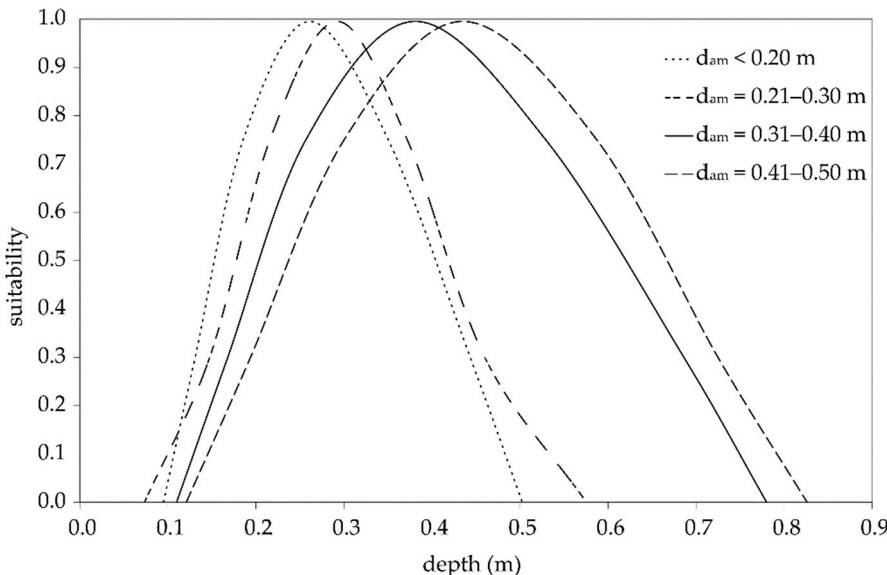

**Figure 2.** Average depth habitat suitability curves (HSCs) for the brown trout divided by the intervals of the average maximum depth [25].

### 2.4. Correlation and Regression Analysis of Instream Characteristics and Their Influence on the Quality of Aquatic Habitat

The main goal was to characterise the interaction of stream characteristics on the quality of the aquatic habitat of mountain and piedmont streams. Table 1 describes the basic statistics on the set of characteristics of the 59 reference reaches. Complete characteristics are given in Supplementary Table S1. The Pearson product–moment correlation coefficient (*r*) [38] was evaluated between these data.

**Table 1.** Basic statistical characteristics of the data set: 364-day discharge ($Q_{364d}$), catchment area to the last profile of the reference reach ($A_p$), maximum value of the water depth in the reference reach ($d_{max}$), average maximum depth from all cross sections ($d_{am}$), average longitudinal slope of the water level ($i_p$), average number of fish covers (microhabitats) per 100 m of reach ($n_{100}$), fish cover length in the flow direction ($L_{Cov}$), fish cover width ($W_{Cov}$), average value of the area-weighted suitability for the monitored fish covers ($AWS_{Cov}$), area-weighted suitability for the monitored reach ($AWS_{Rch}$), minimum values of monitored parameters (Min), variability of the interquartile range using the value of the first quartile (1QR), middle value of the range of monitored data (Median), average values of the monitored parameter (Mean), variability of the interquartile range using the value of the third quartile (3QR), and maximum values of monitored parameters (Max).

| Stream Characteristics | Min | 1QR | Median | Mean | 3QR | Max |
|---|---|---|---|---|---|---|
| $Q_{364d}$ (m$^3$·s$^{-1}$) | 0.003 | 0.025 | 0.052 | 0.084 | 0.095 | 0.43 |
| $A_p$ (km$^2$) | 3.53 | 16.27 | 39.39 | 64.18 | 76.26 | 467.16 |
| $d_{max}$ (m) | 0.1 | 0.17 | 0.27 | 0.302 | 0.41 | 0.79 |
| $d_{am}$ (m) | 0.08 | 0.145 | 0.2 | 0.232 | 0.295 | 0.54 |
| $i_p$ | 0.002 | 0.009 | 0.015 | 0.018 | 0.022 | 0.076 |
| $n_{100}$ | 0.8 | 2.526 | 3.226 | 3.404 | 4.211 | 7.576 |
| $L_{Cov}$ (m) | 5 | 11 | 14 | 15.4 | 18.1 | 32.5 |
| $W_{Cov}$ (m) | 0.35 | 1.05 | 1.4 | 1.826 | 2.12 | 5.84 |
| $AWS_{Cov}$ (m$^2$·m$^{-1}$) | 0.006 | 0.436 | 0.854 | 1.84 | 1.4 | 5.091 |
| $AWS_{Rch}$ (m$^2$·m$^{-1}$) | 0.002 | 0.191 | 0.382 | 0.506 | 0.572 | 2.7 |

M-day discharge ($Q_{Md}$) is the average daily discharge reached or exceeded by M days a year. The values of $Q_{364d}$ for each reference reach were determined in cooperation with the Slovak Hydrometeorological Institute. Actual hydrometric, topographic, and ichthyological measurements were performed in the summer at low water levels ranging from $Q_{90d}$ to $Q_{355d}$.

In ecohydraulics, a habitat can be numerically expressed by a special index named the area-weighted suitability (AWS) in units of m$^2$·m$^{-1}$ and can be calculated as the combined habitat suitability index (CSI) weighted by the area of the water level [23]. The average value of the AWS for the monitored fish covers ($AWS_{Cov}$) was determined according to Equation (1):

$$AWS_{Cov} = \frac{\sum_n^1 (A_{Cov1-n} \times P_{d1-n})}{\sum_n^1 L_{Cov1-n}}, \tag{1}$$

where $A_{Cov}$ = area of the fish cover (m$^2$), $P_d$ = depth suitability determined from the HSC, and $L_{Cov}$ is the fish cover length in the flow direction

The resulting value determines the area (m$^2$) usable for the monitored bioindicator (brown trout) that falls within a microhabitat length of one meter. The $AWS_{Cov}$ value describes the quality of the aquatic habitat at the microhabitat level.

On the other hand, $AWS_{Rch}$ is the value of the area-weighted suitability for the stream within the whole reference reach. This area was determined according to Equation (2):

$$AWS_{Rch} = \frac{\sum_n^1 (A_{Rch1-n} \times P_{d1-n})}{\sum_n^1 L_{Rch}}, \tag{2}$$

where $L_{Rch}$ = reach length in the flow direction and $A_{Rch}$ = total area of the reference reach (m$^2$).

## 3. Results

### 3.1. Habitat Quality

During low-flow periods, the ichthyofauna is concentrated in fish covers; therefore, the number, characteristics, and distribution of fish covers are decisive for evaluation of the quality of the aquatic habitat. As an example, we present the results from the reference reach of the Drietomica stream. Nineteen fish covers were identified within a length of 240 m. At the minimum flow Q = 0.03 $m^3 \cdot s^{-1}$, the total flooded area of the reach was 268.3 $m^2$, of which the fish cover area was 86.21 $m^2$ (32%). The water surface area outside the fish covers was designated as free water. As the flow increased, the total area also increased; at flow Q = 0.08 $m^3 \cdot s^{-1}$, the percentage of the fish cover area remained at approximately 30% of the total surface area. The ichthyological survey, which was carried out at two different flow rates (Q = 0.55 $m^3 \cdot s^{-1}$ and Q = 1.48 $m^3 \cdot s^{-1}$), shows that 93% of brown trout individuals were captured in fish covers [26]. There is a logical assumption that, during periods of low flow, trout prefers fish covers and that the rest of the stream is used for migration. Therefore, it is useful to evaluate the quality of fish covers. The evaluation of the free water is only significant in terms of the possibility of migration (whether the fish are able to migrate through the free water area). Evaluation of 59 reaches shows that the average fish cover area is 36.31% of the total water surface. Figure 3 describes the development of the number of streams with the ratio of microhabitat areas to the total flooded area. The reference reaches of the five streams with low percentages of fish cover areas are predominantly riffle types.

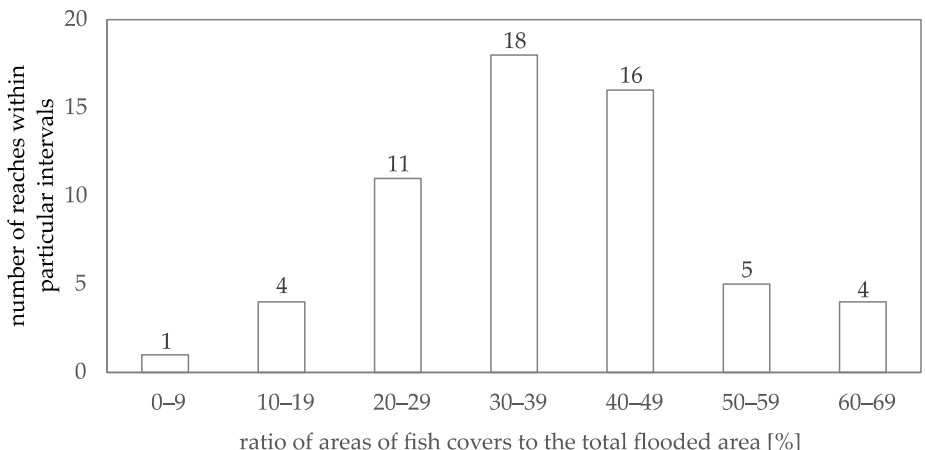

**Figure 3.** Histogram of the number of streams with the ratio of areas of fish covers to the total flooded area.

### 3.2. Correlation and Regression Analyses of Instream Characteristics and Their Influence on the Quality of Aquatic Habitat

Figure 4 shows 10 variables, of which $AWS_{Cov}$ and $AWS_{Rch}$ have been determined to be dependent on other parameters. On the diagonal of the image, there is a name of the variable along with a thumbnail of its histogram. Above the diagonal, there are Pearson product–moment correlation coefficients (*r*) [38] distinguished by colour as negative or positive, and the size of the correlation is emphasized by the font size. Below the diagonal, there are thumbnails of scatter plots with correlation lines. The corresponding correlation coefficient or scatter plot between the two variables can be found at the intersection of the horizontal and vertical from these variables from its box on the diagonal.

Figure 4 shows that $AWS_{Cov}$ has the closest dependence on average width of the fish cover ($W_{Cov}$) and average maximum depth ($d_{am}$). The most significant dependencies are shown in Figure 5 in more detail.

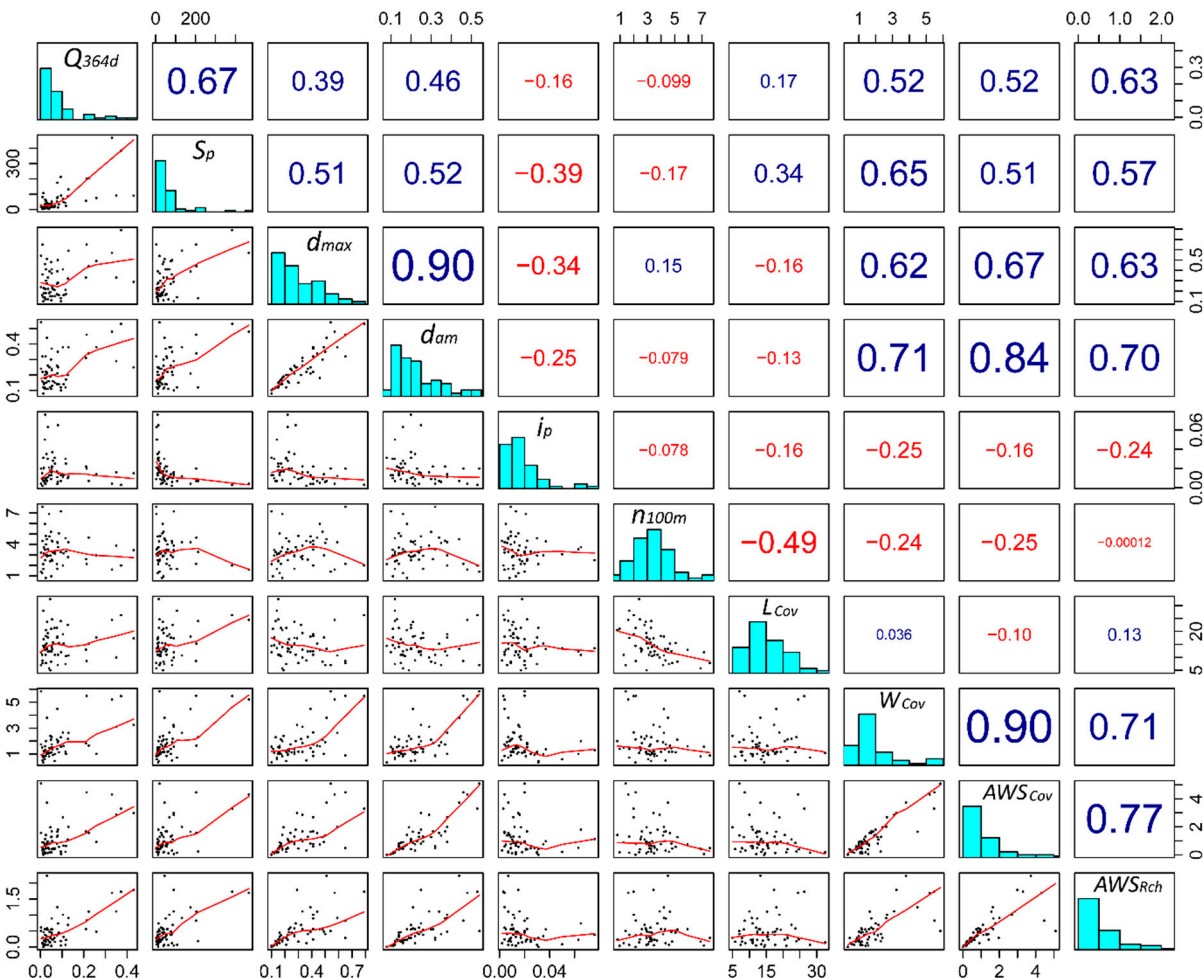

**Figure 4.** Correlation matrix between microhabitat properties and the value of the area-weighted suitability for the monitored fish covers and reaches ($AWS_{Cov}$ and $AWS_{Rch}$, respectively). On the vertical and horizontal axis, there are scales for each individual parameter in accordance with Table 1. Each scale corresponds to a parameter range from Table 1.

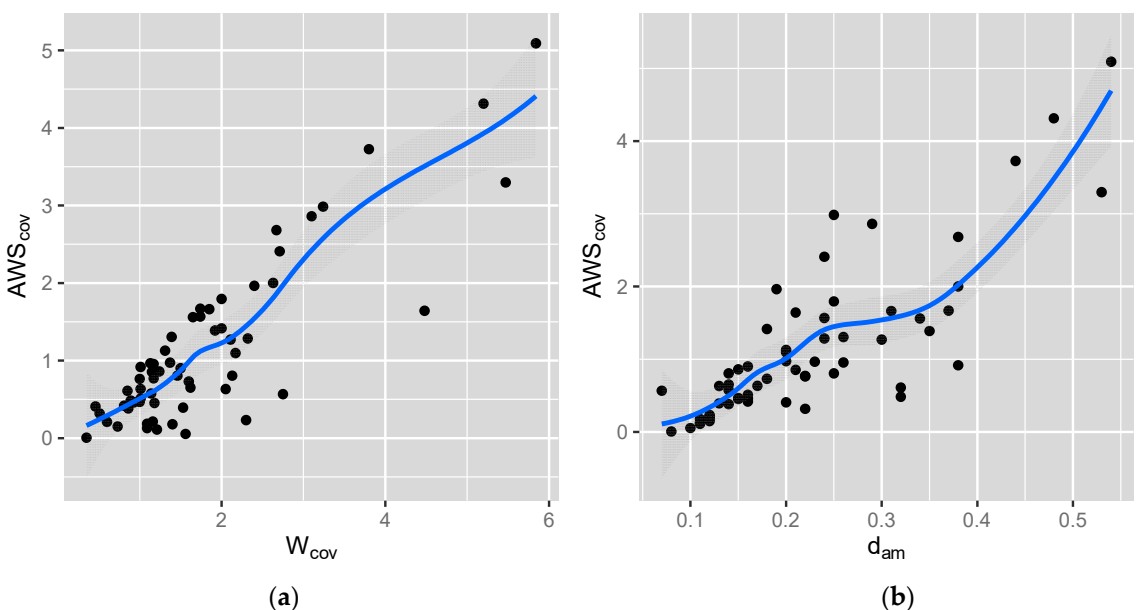

**Figure 5.** The closest correlation relationships for localities within the reference reaches between the value of the $AWS_{Cov}$, and (**a**) fish cover width ($W_{Cov}$) and (**b**) average maximum depth from all cross sections ($d_{am}$).

Table 2 describes the Pearson product–moment correlation coefficient ($r$) [38] between $AWS_{Cov}$ and the individual stream characteristics, together with the $p$-value for the level of 5% and the upper and lower 95% confidence intervals of the value of $r$. The $p$-value is the probability of obtaining test results at least as extreme as the observed results. The table shows that the closest relationships are between $AWS_{Cov}$, and $W_{Cov}$ and $d_{am}$, respectively.

**Table 2.** The correlation coefficient ($r$) between variable $AWS_{Cov}$ and the stream characteristics, $p$-value, and confidence interval.

| Variable | $r$ | $p$-Value | Confidence Interval |
|---|---|---|---|
| $Q_{364d}$ (m$^3$·s$^{-1}$) | 0.500 | $4.16 \times 10^{-5}$ | 0.283–0.667 |
| $A_p$ (km$^2$) | 0.452 | $2.56 \times 10^{-4}$ | 0.226–0.632 |
| $d_{max}$ (m) | 0.623 | $8.44 \times 10^{-8}$ | 0.440–0.756 |
| $d_{am}$ (m) | 0.821 | $5.33 \times 10^{-16}$ | 0.718–0.889 |
| $i_p$ | $-0.172$ | $1.86 \times 10^{-1}$ | $-0.406$–0.084 |
| $n_{100}$ | $-0.243$ | $5.87 \times 10^{-2}$ | $-0.467$–0.009 |
| $L_{Cov}$ (m) | $-0.023$ | $8.62 \times 10^{-1}$ | $-0.273$–0.230 |
| $W_{Cov}$ (m) | 0.868 | $1.29 \times 10^{-19}$ | 0.789–0.919 |

Analysis of the relationship between $AWS_{Cov}$ and $d_{am}$ shows that water depth has a substantial influence on the development of the aquatic habitat quality. This means that conditions for rheophilic species (brown trout and similar) improve as discharge increases [39]. This fact can be used in habitat modelling using IFIM, which mainly uses depth HSCs [40].

For larger streams, the water depth is favourable for brown trout, even at minimum flows. For small streams, where minimum flows do not create enough water depth for brown trout, microhabitats have unsuitable conditions. Therefore, an analysis of the influence of the stream size on the ratio of $AWS_{Cov}$ and $AWS_{Rch}$ to the total flooded surface area was performed. The reaches were sorted in ascending order by $d_{am}$. From Figure 6, it follows that, for torrent streams ($d_{am}$ from 0.08 to 0.20 m), there is a significant increasing trend of $AWS_{Rch}$ and $AWS_{Cov}$. In the $d_{am}$ range from 0.21 to 0.55 m (Figure 7), we can state that the water depth does not affect the ratio of $AWS$ to the total flooded area. $AWS_{Cov}$ increases slightly, while $AWS_{Rch}$ slightly falls.

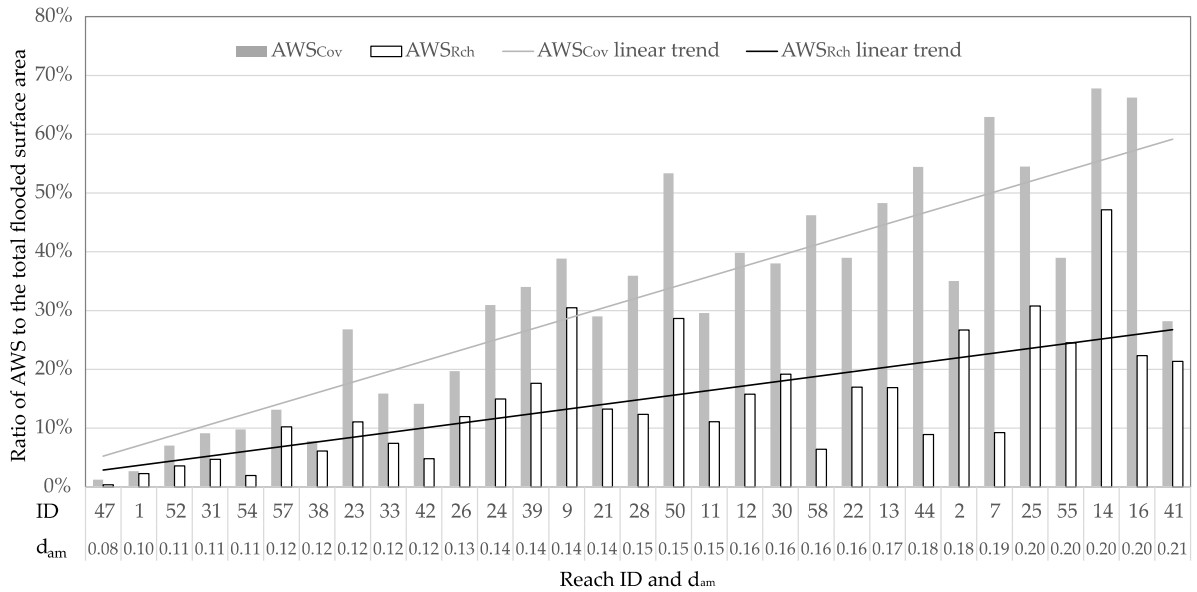

**Figure 6.** Comparison of ratio of $AWS_{Cov}$ and $AWS_{Rch}$ to the total flooded surface area: the reaches are arranged according to $d_{am}$ from 0.08 m to 0.21 m.

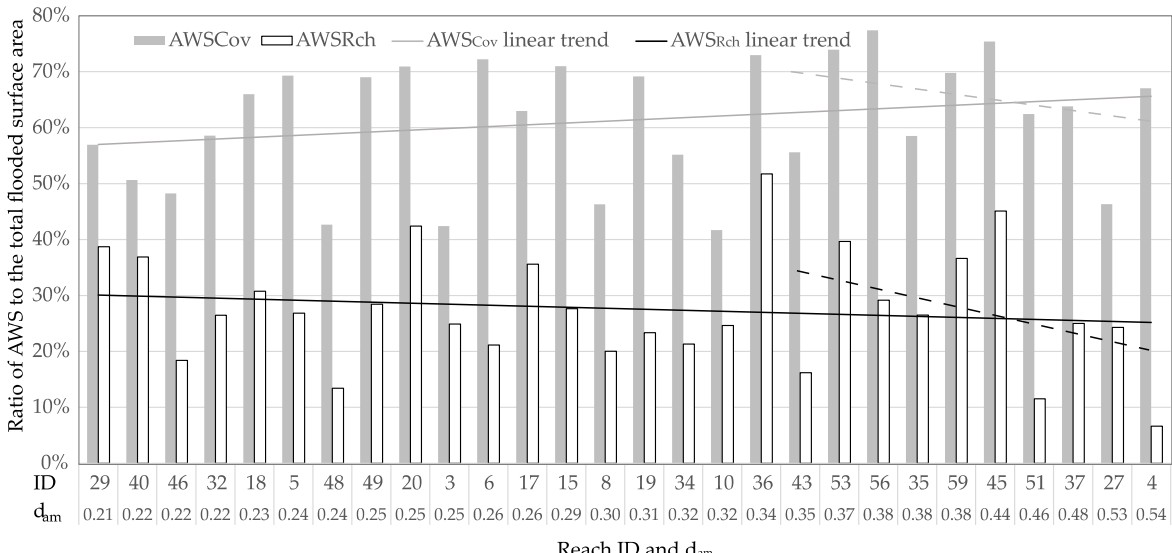

**Figure 7.** Comparison of the ratio of $AWS_{Cov}$ and $AWS_{Rch}$ to the total flooded surface area: the reaches are arranged according to $d_{am}$ from 0.21 m to 0.54 m. For $d_{am}$ from 0.35 m to 0.54 m, the trend line is depicted by the grey dashed line ($AWS_{Cov}$) and the black dashed line ($AWS_{Rch}$).

A similar result was found in the relationship between the values of $d_{am}$ and $W_{Cov}$. The reaches were divided into four intervals according to $d_{am}$. The intervals were chosen according to Figure 2. In the first three intervals (<0.2 m, 0.2–0.3 m, and 0.3–0.4 m) the water level width gradually increased from 1.3 m to 1.7 m. In the last interval (0.4–0.5 m), however, there was a significant change in the average water level width of 5.16 m. The reason for this change may be the transition of the mountain reaches to the piedmont reaches. The change is also in accordance with the results in Figure 7. In the reaches with $d_{am}$ 0.38–0.54 m, there is a significant decrease in the ratio of AWS to the total water surface area.

## 4. Discussion

The results of the correlation analyses in Tables 1 and 2 show the mutual relationship between $AWS_{Cov}$ and $W_{Cov}$, $d_{max}$, $Q_{364d}$, $A_p$, and $d_{am}$. A correlation was not confirmed between the parameters $AWS_{Cov}$ and $i_p$, $n_{100}$, and $L_{Cov}$. The correlation analysis of the $AWS_{Rch}$ variable was also performed; however, the resulting correlation values were lower than those for $AWS_{Cov}$. This result implies that the quality of the aquatic habitat for brown trout is primarily determined by the cover possibilities of the stream. The rest of the stream has little effect on trout when this part of the stream does not create migration barriers. It can be stated that the impact of technical interventions, such as river regulation, can be determined according to the water depth conditions, based on stream covers with the appropriate water depth and flow velocity [26]. A bioindicator that is sensitive to morphological changes, such as trout, must be selected. Ecohydraulic research has confirmed the dominant effect of flow velocity and water depth on fish habitats [41]. This fact is also reflected in the modelling of aquatic habitat quality using the IFIM methodology. Studies have confirmed a correlation between the characteristics of the HSCs and hydraulic parameters, especially the depth and velocity [25,42]. River regulation also changes the velocity field of the stream, which has a minor effect on habitat quality during periods of minimum flow. This was also confirmed by the results of our previous research, where the optimal ratio of the influence of velocity and depth, expressed in the form of HSCs, was 2:8 [43].

Analysis of the influence of the stream size on the ratio of $AWS_{Cov}$ and $AWS_{Rch}$ to the total flooded surface area demonstrated that, in torrent streams, there was a gradual increase in the $AWS$ ratio to the total surface area (streams up to $d_{am}$ = 0.20 m). The

piedmont streams were not sensitive to changes in water depth, even at greater depths, and there was a reduction in the *AWS* share. The reason for this development may be that, in shallow streams, the area of optimal habitats (with sufficient water depth) is low. By increasing the discharge, the habitat area is higher (approximately 40% or more of the entire surface area). It can be stated that, the larger the stream, the less sensitive it is to changes in depth and that there are minimal changes in the area of suitable habitats with changes in depth. In deeper streams (in our case, 0.4–0.5 m), there may be a decrease in the *AWS* with increasing water depth. This trend is likely to be affected by a change in the velocity field. This means that there are also higher flow velocities at greater depths, which can have a negative effect on ichthyofauna habitat preferences. In other words, changes in the morphology of the stream bed, primarily the depth conditions induced by river regulation, also affect the velocity field of the stream. Water quality does not radically change with river regulation. It is logical that the aquatic habitat quality not only of the regulated stream but also of the restored stream should be evaluated particularly based on stream morphology and bioindicator, in this case, ichthyofauna. Such an evaluation is provided by models based on the IFIM methodology, such as SEFA. Modelling a larger set of streams using these models would be extremely challenging; in the case of classification of water bodies into five classes, as required by the Water Framework Directive [22], it would be nearly impossible. From the above statistical analysis, in order to interpret the relationships between the dependent variable $AWS_{Rch}$ and other variables, it follows that 5 variables that result from the stream characteristics are important. The characteristics are $Q_{364d}$ ($m^3 \cdot s^{-1}$), $A_p$ ($km^2$), $d_{max}$ (m), $d_{am}$ (m), and $W_{Cov}$ (m). Figure 8 shows the relative importance of these variables, as determined by the method published in [44].

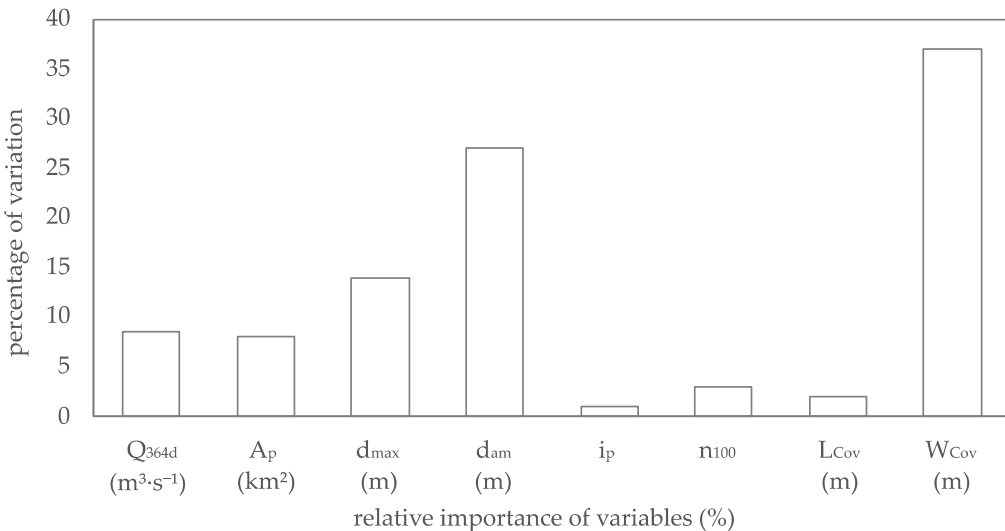

**Figure 8.** Relative importance of variables (%).

These results are the basis for the creation of regression equations which can be used to determine the dependent variable of $AWS_{Rch}$. In practice, this will mean that, instead of a detailed topographic survey, hydraulic modelling, and ichthyological research, $AWS_{Rch}$ will be set based on average values of $A_p$, $d_{max}$, $d_{am}$, and $W_{Cov}$, which can be determined during field reconnaissance of the stream and the discharge to be hydrometrically measured. Of course, generalized HSCs of the representative species must be available. Such a procedure can significantly simplify evaluation of the quality of aquatic habitat, making it much more accessible for design practice.

The database for deriving regression equations was divided into calibration data (59 reaches representing the data evaluated by the authors for 22 years of field measurements) and validation data (for which other measurements are currently being performed). The regression results of the calibration file show a high value of the coefficient of determi-

nation ($r^2 = 0.8452$). This determines the ratio of the common variance of the independent and dependent variables, i.e., how much the change in independent variables affects the $AWS_{Rch}$ dependent variable. These results also suggest that determination of $AWS_{Rch}$ by regression equations is promising.

## 5. Conclusions

The results of this research in the reference reaches of mountain and piedmont streams in Slovakia indicates that the quality of instream habitats can be characterized by the relationship between the morphological characteristics of the stream and the biotic characteristics represented by the *AWS*. The results of the correlation analysis demonstrate the influence of individual parameters on the instream biota. Furthermore, brown trout respond sensitively to abiotic parameters, such as riverbed morphology, flow velocity, and water depth. Based on the obtained results, it is possible to determine the characteristics of the stream which have a substantial effect on the instream habitat quality. Specifically, based on the morphological changes, which are represented by habitat suitability curves for water depth, it is possible to evaluate the impact of river regulation or to predict the effect of stream restoration.

**Supplementary Materials:** The following are available online at https://www.mdpi.com/2073-4441/13/2/142/s1, Table S1: Basic characteristics of the data set, Figure S1–S5: Photos from the reaches taken during field surveys to illustrate the character of the reaches.

**Author Contributions:** Conceptualization, V.M. and P.I.; methodology, V.M., M.Č. and P.I.; software, M.Č., P.I. and A.Š.; validation, Z.Š., V.M. and A.Š.; formal analysis, A.Š., M.M., V.T. and Z.Š.; investigation, V.M., P.I. and A.Š.; resources, V.M. and A.Š.; data curation, V.M.; writing—original draft preparation, Z.Š. and V.M.; writing—review and editing, A.Š., M.M., V.T. and Z.Š.; visualization, P.I., M.Č. and A.Š.; supervision, V.M. and A.Š.; project administration, V.M. and A.Š.; funding acquisition, V.M. and A.Š. All authors have read and agreed to the published version of the manuscript.

**Funding:** This research was jointly funded by the Slovak Scientific Grant Agency, grant No. VEGA 1/0068/19; by the Slovak Research and Development Agency, grant No. APVV-16-0253; and by the Ministry of Education, Science, Research, and Sport of the Slovak Republic within the Research and Development Operational Programme for the project "University Science Park of STU Bratislava", ITMS 26240220084, co-funded by the European Regional Development Fund.

**Institutional Review Board Statement:** Following the legislation of Slovak Republic, the liability to notify the Slovak Fishery Union (SFU) as the responsible organization was fulfilled one week prior to each sampling. A representative of SFU was present at each sampling to supervise the correctness of the methods used. Each sampling was performed by a person with special permit for electrofishing issued by the Ministry of Environment of the Slovak Republic. All the fish samples collected by electrofishing were immediately counted, weighted, measured, and returned back to original locality in the stream. All sampling procedures used in this study strictly followed the ethical standards and animal welfare declared during the process of obtaining the field permit from the Ministry of Environment of the Slovak Republic. Authors declare that all customary standards concerning handling the live material applicable in the EU were complied.

**Informed Consent Statement:** Not applicable.

**Data Availability Statement:** The data presented in this study are available within the article and Supplementary Material.

**Conflicts of Interest:** The authors declare no conflict of interest. The funders had no role in the design of the study; in the collection, analyses, or interpretation of data; in the writing of the manuscript; or in the decision to publish the results.

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
