# Peer review of "Relationship between Morphological Characteristics and Quality of Aquatic Habitat in Mountain Streams of Slovakia"

_water, doi:10.3390/w13020142_

Round 1
Reviewer 1 Report
This paper examines the relationship between geomorphic parameters and fish habitat quality in a mountain and piedmont river in its natural state. The relationship between habitat quality and geomorphic parameters is valuable as a report because it can provide guidance for eco-friendly river regulation and nature restoration projects. However, I couldn't understand the novelty of this paper because there have been many studies, mainly in the field of ecology, that have linked freshwater fishes (including brown trout) to riverine geomorphological parameters. This paper is limited to a review from the hydrological aspect. In addition, the methodology is often described as following ISO, and it is not possible to evaluate the methods used to carry out the studies.
A major revision is necessary so as to clarify the purpose and uniqueness of the paper and adequately describe the methods used for that purpose.
Reviewer 2 Report
General Comments:
The study is interesting and evaluates the relationship between morphological characteristics and aquatic habitat suitability using Instream Flow Incremental Methodology (IFIM). The findings highlighted the influence of bed morphology, flow velocity, and water depth on the stream biota.
However, my personal opinion to the manuscript is a major revision with some additions and modifications. While doing the revision, the authors should consider the following points:
- The introduction needs some improvements. The authors need to clearly state the motivation for this study.
- The authors also need to clearly state the novelty of the work.
- In my experience, measurement of shallow flows where roughness elements are of similar size to depth is difficult. Could the authors elaborate and give estimates of probable topographic and hydrometric measurement error?
- The presentation of this manuscript would be enhanced with several photos of the sites.
Specific Comments:
- Lines 83-84, is the objective, can be moved at the end of the introduction.
- Line 106, “hydraulic model” needs further clarification.
- In Conclusion, the authors may include some insight for future research.
Reviewer 3 Report
General comments:
Štefunková et al. studied the relationship between abiotic flow characteristics and habitat quality. Fifty-nine references reaches of fifty-two mountain and piedmont streams in Slovakia was analyzed. The authors used Instream Flow Incremental Methodology where brown trout was chosen as bioindicator. The study is interesting and current especially because of huge number of river regulation which impacting on quality of aquatic habitat and biota diversity. On the other hand I have some problem with finding a novel problem…on this paper. In my opinion the general conclusions are too popular and we can find huge number of papers with similar concept. Another problem is also local rang of the studied area. There are some obscurity in the manuscript. My recommendation for the current submission is to Major revision. A list of comments with line numbers is given below.
The reviewer comments:
-The title
Line 2-4 The title in my opinion is not correct. I propose to add the locality of the streams at the end of the title, e.g. Relationship between morphological characteristics and quality of aquatic habitat of mountain streams in Slovakia.
- Abstract and Keywords
Line 27-28 General the abstract is right, but the last sentence is too obvious. When you prepare the conclusions please be more precise.
- Introduction
The introduction is general proper, but needs some corrections.
Line 37-39 This sentence is too obvious, we know that for a long, long time…
Line 77-79 Really ?, so what about fish as a bioindicators in regulated reaches of rivers? Do this animal can show precisely the habitat quality in case that they have some problems with e.g. migration (especially when some damming or hydrotechnical structures are used as a regulation). And what does it mean “traditionally” ?
Line 80-81 This aim is not proper, it is too general… What are the research hypotheses? This part of your paper is one of the most important so in this form is not acceptable. Please improve it.
- Material and Methods
Line 83-85 Is it the aim of the study or M&M? I see a mess in this part. I have also some question about the study design: I would like to ask about the length of the each studied reaches? How many times the fish have been measured? Did you measured the physicochemical properties? You do not mention about the type of stream regulations, I think it is important information. What kind of microhabitat characteristics do you measured?
Line 126-128 I can’t agree with that. In the literature the are a lot of paper which confirm that macroinvertebrates are a sensitive indicator of river regulations. Do you have an information that they should not be use in such study? Please give some references confirming your statement. Also I do not understand why you mention about it if you do not use aquatic invertebrates in analysis…
Moreover please add a description of the statistical analysis to the M&M section of the manuscript.
- Results
Line 181 If you use the stream name maybe you should prepare some appendix with name of studied streams and some general hydrological characteristics.
Line 203 What kind of correlation coefficient did you used?
Line 237-238 Improve and change [cm] to [m].
- Discussion
Discussion is very poor and short, it is shorter than introduction. There is a need to emphasize the obtained results in relation to the literature. Please improve it.
Figure 2 [m] or [cm]...? In the whole of the paper you use [m], please be consistent.
Figure 3 Number of what? Improve the caption of the axis OY.
Figure 6 Improve the caption of the axes, once again you use [cm]
Figure 7 Improve the caption of the axes, once again you use [cm]
Round 2
Reviewer 1 Report
The authors revised appropriately, so I recommend that this paper be accepted.
Reviewer 2 Report
The paper has been sufficiently revised in relation to the previous version and I am satisfied with the newer version of the manuscript.
Reviewer 3 Report
Dear authors,
after reviewing your manuscript improvements, I decided to accept it for publication in present form.